# Effects of Process Parameters and Process Defects on the Flexural Fatigue Life of Ti-6Al-4V Fabricated by Laser Powder Bed Fusion

**DOI:** 10.3390/ma17184548

**Published:** 2024-09-16

**Authors:** Brandon Ramirez, Cristian Banuelos, Alex De La Cruz, Shadman Tahsin Nabil, Edel Arrieta, Lawrence E. Murr, Ryan B. Wicker, Francisco Medina

**Affiliations:** 1Department of Aerospace and Mechanical Engineering, University of Texas at El Paso, El Paso, TX 79968, USA; bramirez16@miners.utep.edu (B.R.);; 2W.M. Keck Center for 3D Innovation, University of Texas at El Paso, El Paso, TX 79968, USA

**Keywords:** Ti-6Al-4V alloy, additive manufacturing, laser powder bed fusion, porosity defects, surface roughness, fatigue performance, Basquin’s law fitting

## Abstract

The fatigue performance of laser powder bed fusion-fabricated Ti-6Al-4V alloy was investigated using four-point bending testing. Specifically, the effects of keyhole and lack-of-fusion porosities along with various surface roughness parameters, were evaluated in the context of pore circularity and size using 2D optical metallography. Surface roughness of S_a_ = 15 to 7 microns was examined by SEM, and the corresponding fatigue performance was found to vary by 10^2^ cycles to failure. The S–N curves for the various defects were also correlated with process window examination in laser beam power–velocity (P–V) space. Basquin’s stress-life relation was well fitted to the experimental S–N curves for various process parameters except keyhole porosity, indicating reduced importance for LPBF-fabricated Ti-6Al-4V alloy components.

## 1. Introduction

Metal fatigue began to be studied systematically in the mid-1800s as distinguished by the failure of parts during dynamic (cyclic) loading at stresses below the stresses at which these same parts failed by static (tensile) loading [1]. During the late 1800s and early 1900s, A. Wohler introduced the concept of fatigue “endurance limit”—a stress below which fatigue failure would not occur [2]. In 1910, O. Basquin [3] showed that alternating stress (S) versus the number (N) of alternating stress cycles to failure (the S–N curve) could be represented by a log–log linear relationship [2].

Over a century of metal fatigue analysis, and especially studies of the mechanism of fatigue failure [4], the role of defects and microstructures on crack initiation and growth were identified. These observations, along with empirical modeling of fatigue data, allowed for the development of design strategies to optimize the application of engineering metals and alloys in fatigue environments. The principal defects contributing to metal fatigue consisted of internal voids and inclusions, which initiated cracks, along with surface roughness-related features consisting of scratch and notch-related phenomena, along with other surface features exaggerated by wear and corrosion, also contributed to crack initiation and growth [4].

In the early part of the current century (2000s), additive manufacturing (AM), utilizing metal and alloy powders melted in layer-by-layer configurations by electron and laser beams, emerged [5]. This new manufacturing paradigm provided an opportunity to fabricate complex products and structures where fatigue became a particularly important aspect of engineering applications of these novel metal and alloy parts. In this regard, it has been observed that the microstructures and mechanical properties, including fatigue behavior, of AM-fabricated metals and alloys can differ significantly from more conventionally manufactured products [5], providing particular challenges for understanding and predicting fatigue life [5,6]. Over the past roughly 2 decades, several thousand publications have described fatigue phenomena and associated data, especially S–N curves, for a range of metals and alloys fabricated by laser powder bed fusion (LPBF) AM [5,7]. A recent paper by Zhang and Xu [8] presents an extensive fatigue database of AM metals and alloys, including stress–life (S–N) data extracted from more than 3000 scientific publications up to 2022, including data fitted using a log-normal Basquin equation [3].

Ti-6Al-4V alloy has been one of the most studied alloys fabricated by AM, especially LPBF [5,9,10]. It is used prominently in many industrial and aerospace applications because of its corrosion resistance, desirable strength-to-weight ratio, and related mechanical properties, including fatigue performance. A wealth of fatigue studies for LPBF-fabricated Ti-6Al-4V have provided significant information involving intrinsic microstructures and defect properties as a function of process parameters over the past two decades [5,8,9,10,11,12,13]. This has included studies of microstructures and other defects, such as porosity and surface roughness features unique to AM processing involving layer-by-layer powder fabrication, especially pertinent to fatigue performance [14,15,16,17,18,19,20,21,22,23,24,25,26,27,28,29,30]. Wang, et al. [28] have noted that porosity in LPBF-fabricated alloys such as Ti-6Al-4V can be described in terms of quantity, size, morphology, and location, especially in relation to the component surface. These porosity issues can arise from gas bubbles originating in the precursor powder and the inability to adequately melt the powder layers (lack of fusion) due to inadequate processing parameters, especially laser beam power (P) and beam scan velocity (V) [5]. Varying degrees of unmelted powder on the surface of LPBF-fabricated products also pose detrimental features as a result of variations in surface roughness features [18,23,24]. While much of any subsurface porosity can be reduced or eliminated by heat treatment, especially HIP [19,21,22,31,32,33,34], and surface roughness correspondingly reduced by post-processing as well as post-process machining or chemical processing of the surface [33,34,35], the fabrication and application of complex, AM (LPBF) parts often requires maintaining the as-built condition. Additionally, Foti, et al. [36] have noted that complex, AM-fabricated parts or systems often experience multiaxial (flexural) fatigue. Correspondingly, Ariza, et al. [37] have also recently compared the four-point and uniaxial fatigue performance for L-PBF-fabricated Ti-6Al-4V alloy. Rogers, et al. [38] recently described methodologies for quantifying AM fatigue data uncertainty.

The effects of internal defects—lack of fusion (LoF) porosity, keyhole porosity (entrapped gas bubbles) and their sizes, shapes and density, along with various surface roughness features on fatigue performance—have been extensively studied for uniaxial fatigue of conventionally processed metals and alloys [1,2,4], as well as AM-fabricated metals and alloys [5,6,7,8,11,12,13,14]. Correspondingly, four-point bending (multiaxial) fatigue has not been as extensively studied in spite of the fact that many metal and alloy products and system configurations experience multiaxial loading. And, as noted above, there have been numerous studies evaluating porosity and surface roughness issues influenced by processing parameters such as laser beam power (P) (and power density), scan speed (V), beam size, hatch spacing, and layer thickness on fatigue performance, there is a dearth of studies establishing the relationship connections between as-built component process parameters, residual defects and fatigue crack formation and propagation on fatigue life. Greitemeier, et al. [39] recently showed that the fatigue properties of AM-fabricated Ti-6Al-4V are dominated by surface roughness effects. Lee, et al. [25] have also argued that conventional surface roughness parameters—S_a_, the mean arithmetical roughness feature height; S_v_, the maximum valley depth profile roughness; and S_vk_, the average pit depth [40]—do not adequately represent the correlation between surface roughness and fatigue life.

Recent reviews by Cao, et al. [16] and Sanaei and Fatemi [18] begin to address these correlations and their connections to AM process parameters and fatigue life, while a more recent study by Narra, et al. [23] begins to address this shortcoming by developing a process qualification approach using four-point fatigue testing of LPBF-fabricated Ti-6Al-4V alloy to illustrate the connections between processing-defect structure-fatigue life using a process window or process map concept. This concept was based on previous studies by Gordon, et al. [41], Read, et al. [42], and Islam, et al. [43]. This approach began to provide an overview of process qualification to establish a process-defect structure-fatigue life relationship for LPBF of Ti-6Al-4V products. Narra, et al. [23] found that Ti64 LPBF fabricated in the center of the process (power–velocity) window showed the best fatigue life performance, while lack-of-fusion (LoF) conditions resulted in the worst fatigue performance.

The present paper provides an extension to the recent work of Narra, et al. [23] on LPBF-fabricated Ti64 alloy while also complementing the prior work of Cao, et al. [16]. This research involved the comparison of four-point bending (flexural) fatigue testing of LPBF-fabricated Ti64 alloy using a simplified power–velocity (P–V) map corresponding to 30-micron layer thickness as a guideline to print within the keyholing region, the optimized process window, and the lack-of-fusion (LoF) region. Of these three regions, five different laser power levels and corresponding beam scan speeds were selected and randomly assigned to 45 specimens. In addition, the effect of surface roughness (including S_a_, S_v_, and S_vk_) implicit in surface particle melt features on fatigue performance was also investigated and compared with the porosity-related fatigue performance issues and process parameters by plotting the corresponding peak fatigue stress (S) against the number of cycles to failure (N) in S–N diagrams; and these results were compared with those of Cao, et al. [16] and Narra, et al. [23]. Another interesting feature of fatigue life investigated in this study involved the application of Basquin’s law (which, noted above, is more than 100 years old [3]) to fit the experimental S–N curves for specific process parameters and defects. Additionally, the variance of the fitted Basquin S–N curves was also compared with the porosity and surface roughness parameter data.

## 2. Background

It has been generally shown that powder-based additive manufacturing processes are associated with several as-built component defects aside from microstructures, such as phase or grain size and morphology [9,10,11,14,15,16,17,18,19,20,23]. These process-specific defects, which have an overriding effect on fatigue performance, include lack-of-fusion (LoF) porosity, keyhole porosity, especially entrapped gas, and surface roughness-related stress concentration features. The size, shape, and density of pores and void regions specially create stress state/strain localization regimes that affect fatigue crack initiation and subsequent fatigue endurance limits [14,15,16,17,18,19,20,23,24,25,26,27,28,29,30].

### 2.1. Lack of Fusion Defects

Lack of fusion is a defect in LPBF where poor laser penetration on the melt pool does not allow proper fusion between adjacent layers [12,31,32]. The reasons for poor laser penetration can occur for several reasons but are closely related to the power and hatch spacing of the laser beam and the speed at which the beam scans [33]. The changing of these variables alters the overall size of the melt pool, and decreasing the size of the melt pool makes parts more vulnerable to a lack of fusion defects. This lack of binding within subsequent layers due to LoF leads to an increase in porosity, affecting the overall mechanical performance of the component being built [44,45]. Although the defects may not always be seen on a macroscopic scale, the defects are apparent under a microscope. Methods for mitigating the occurrence of LoF pores exist through non-destructive testing, such as CT scanning, and newer methods, such as seen by Sam Coeck et al. [46], where the light emitted from the laser spot can be recorded to predict LoF porosity.

### 2.2. Keyhole Defects

In contrast to LoF, keyholing occurs when the laser power is too high or when the scanning velocity is not fast enough to dissipate heat. Keyhole defects are typically spherical and are also associated with the entrapment of gas in the melt pool during the printing process. The gas can vary between the inert gas used when printing or gas bubbles in the metal when atomized. Keyhole porosity also contributes to premature mechanical failure within components.

### 2.3. Process Window Parameters

The process window parameters are parameter sets that have a higher possibility to increase the density within printed components, limiting the occurrence of keyhole and lack of fusion defects. The highest-density parts are printed with these parameter sets; however, there are variances within the surface finish of the as-built surface. This, in turn, will have structurally sound products; however, they will be weakened due to the stress concentrations at a rough surface. For this reason, many parts undergo surface post-processing to reduce the number of stress concentrations.

### 2.4. Surface Roughness

The surface integrity of a component can have a significant impact on cyclical life. The surface integrity can be affected by surface defects, surface roughness, and surface porosity. Imperfections within a surface allow for stress concentrations to be present, which accelerate the rate at which a crack forms and propagates [45]. Defects along the surface are initiations of microcracks, which in turn become microcracks, leading to premature failure of the component [47]. On the contrary, a smooth surface with minimal defects is expected to have a higher cyclical (fatigue) life because of the mitigation of surface defects [48]. There is a benefit to surface post-processing methods; however, this is not without increasing the cost and lead time of manufacturing [49]. As noted earlier, principal surface parameters involve S_a_, S_v_, and S_vk_, discussed in detail by He, et al. [40] as these apply to engineering applications.

## 3. Materials and Methods

### 3.1. Four-Point Bending Fatigue

While fatigue testing is pertinent to assess the mechanical behavior of materials, there is an abundance of methods used to assess cyclical life. For example, the effects of surface finishing methods, the effects of stress concentrations, crack nucleation, and crack propagation are data relevant to fully understanding the material behavior [50]. However, to fully understand the mechanical behavior of a material, it is necessary to conduct various types of cyclical testing.

Among some of today’s common practices for assessing cyclical life are uniaxial tension/compression, beam bending, and indirect tensile testing [51]. Specifically, for additively manufactured Ti64, there is a plethora of uniaxial and three-point bending literature that can be found; however, there does seem to be a lack of four-point bending literature. Although three-point bending and four-point bending are very similar because the flexural cyclical life of the beam is examined, the addition of an extra contact point on a specimen in four-point bending alters the loading conditions during testing. An illustration can be seen in Figure 1, where there is a side-by-side comparison between the shear-moment diagrams of three-point and four-point bending. In Figure 1, *F* signifies the force on the specimen for each support, *L* is the length of the sample, and *s* is the location measured from the left-most support. There is a difference between the point where the maximum stress is applied on the specimen because four-point bending creates a region of maximum stress versus a point as seen in three-point bending. The benefit of four-point bending is that the probability of fracture occurring within the most stressed region is increased.

### 3.2. Powder Feedstock and Printing

The powder used for the LPBF samples was gas-atomized Ti-6Al-4V powder provided by ATI (Markham, ON, Canada). Table 1 depicts the chemistry composition of the powder feedstock. The particle size distribution of the Titanium powder is D90 = 57.4 µm, D50 = 37.2 µm, D10 = 26.4 µm.

The titanium samples were printed using an EOS M290 (EOS GmbH, Krailling, Germany), a German-made LPBF printer with a 400 W Ytterbium fiber laser. The build plate dimensions of the EOS M290 are 250 mm × 250 mm × 325 mm.

### 3.3. Printing Parameters

Five process parameters were defined within the power–velocity (P–V) map at 30 µm layer thickness to be randomly assigned to the samples. The printing regions within a PV map have been determined in the literature through experimentation, as shown in Figure 2; for example, in the article by Jerard V. Gordon et al. [41], various scanning speeds and laser power combinations were examined to analyze defect formation with Ti64 printed specimens. After ensuring the parameters determined were within each of the three main regions of a PV map, each sample was randomly assigned one of the five defined process parameters and then printed. A total of 45 [5 × 5 × 70] mm rectangular samples were printed in the diamond orientation shown in Figure 3. The specific laser power and scanning velocity can be shown in Table 2. To ensure residual stresses did not affect the fatigue life of the samples, the forty-five samples were stress-relieved at 600 °C for 120 min in a vacuum.

### 3.4. Scanning Strategies

It is shown that EOS nominal (L_p_ = 280 J, V = 1200 mm/s) and EOS nominal Improved (L_p_ = 280 J, V = 1200 mm/s) have the same process parameters; however, they differ in the scanning strategy used during printing. Figure 4 shows the workflow between EOS nominal and all the other parameters. The key differences between the two scanning strategies are the hatch offsets, contour offsets, and the number of contours. Contour 3 was used to remelt the prior printed contours to create a less rough surface finish.

The sample process parameters were sliced using EOS Print. The different vector paths can be seen in Figure 5 where the difference in contour offsets can be seen between EOS nominal and all other process parameters are shown. The values of the parameters used are shown in Table 3 and Table 4.

### 3.5. Fatigue Testing

The testing apparatus consisted of a four-point bending fixture manufactured by Material Testing Technologies Inc. (Wheeling, IL, USA), an MTS Landmark, an alignment fixture, and the Multipurpose Elite (MPE) software package by MTS. Using the MPE software, a procedure was programmed to ensure all testing conditions remained constant while allowing for key parameters such as stress levels to be altered.

Using the four-point bending fixture and the MTS Landmark, the samples were placed as shown in Figure 6. An alignment tool was designed and manufactured through material extrusion to maintain proper alignment when placing the specimens between the four supports. The distance between the top two supports is 10 mm, and the distance between the bottom supports is 30 mm. Since all samples are 70 mm in length, one sample undergoes two tests at two different stress levels. The stress ratio was defined to be R = 0.1; the minimum testing stress was 10% of the maximum stress. The speed of testing was conducted at 10 Hz and if a sample were to fracture during a cycle, displacement and force limits were defined on the machine to prevent a crash between the load cell and the crosshead.

### 3.6. Porosity Analysis

Images of the mounted specimens were recorded with a Keyence VHX-7000 optical microscope (Keyence, Mechelen, Belgium). Image J 1.54, an image processing tool by NIH, was then employed to evaluate the number of pores, their size, and shape fidelity. Adjustments in the image hue, saturation, and brightness heightened the contrast for the pores, enabling Image J detection. The software could autonomously tally all core porosity and fringe porosity, the last of which pertains to pores within 300 microns of the interior. Using the circularity function of Image J,
Circularity = 4*π* (area)/perimeter^2^, (1)
the shape regularity of the pores was determined. These assessments produced values on a spectrum from 0 to 1, where 1 indicated perfect circularity. An aggregate circularity value was subsequently determined for each process parameter by averaging individual pore values.

## 4. Results and Discussion

### 4.1. Internal Defect (Porosity) Analysis

A cross-sectional analysis was performed on all process parameters shown in Table 2 and Figure 2 and described in Section 3.4 above. Figure 7, Figure 8 and Figure 9 illustrate XY cross-section images recorded as discussed in Section 3.5 above. Density measurements were also determined using a Micrometrics AccuPyc II 1340 gas pycnometer and by using the 2D image cross-sections and converting them to percent porosity. Image J and the Keyence system (Section 3.5) were used to analyze the XY cross-sections illustrated in Figure 7, Figure 8 and Figure 9 to find the number of pores per cross-section, their average size (diameter), and average circularity (Equation (1)). It is well documented that pores having very irregular shape, resulting in low circularity, have a much greater influence on fatigue performance than more circular pores with a high circularity as a consequence of a higher propensity for crack initiation at irregular pores [26,28,52,53]. The keyhole and LoF parameters induced many internal defects with an average circularity of 0.76 and 0.63, respectively, consistent with the porosity shape variances shown typically in Figure 7 and Figure 9. In contrast, Figure 8 shows almost no porosity for the EOS normal printing parameter. The keyhole and LoF process parameters (Figure 7 and Figure 9) represented the largest number of internal defects compared to the process parameters printed within the process window as well as the EOS nominal and nominal improved printing parameters. These results are summarized for comparison in Table 5.

Although three parameters were printed within the process window regime, there were significantly more defects for the P5 process parameter when compared to the EOS nominal and nominal improved parameters. In addition, the density of P5 is comparable to the EOS nominal and nominal improved process parameters shown in Table 5.

### 4.2. Surface Roughness Analysis and Comparison

When comparing the average surface roughness of EOS nominal (NOM) parameters to EOS nominal improved (NOM IM) parameters using a Keyence VHX 5000, it was observed that EOS NOM parameters had an average surface roughness of S_a_ = 15 µm while the EOS Nominal Improved (NOM IM) surface roughness values were S_a_ = 7 µm; more than a factor of 2 less. Figure 10 compares the SEM images of the two surfaces. These results strongly suggest that the scanning strategies have a significant impact on the surface roughness despite having the same process parameters for printing.

Using the Keyence VR-5000, the surface roughness for each process parameter was taken along the side of the specimen undergoing tension. The overall surface roughness between all process parameters varied, similar to those observed in Figure 10. For example, the keyhole specimens (P3 in Figure 2) were observed to have a S_a_ = 12 microns in contrast to the process window specimens (P5 in Table 2 and Figure 2), and the LOF specimens (P8 in Table 2 and Figure 2) with S_a_ = 9 microns and 8 microns, respectively. These results are shown in Table 6, which includes all measured surface roughness parameters. It is notable that for all surface roughness parameter measurements, the EOS nominal (Figure 10) represents the highest value for all measured parameters: average surface roughness, S_a_, maximum valley depth, S_v_, and average valley depth, S_vk_. These parameters each track the same variance for the process parameters, where S_a_ and S_v_ represent nearly identical values. These results confirm the general reliance on average surface roughness (S_a_) measurements to represent the overall concept of surface roughness, especially as it applies to LPBF Ti-6Al-4V fabrication process (printing) parameters.

### 4.3. Fatigue Performance Analysis and Comparison

Fatigue testing of all specimens represented in the P–V map of Figure 2 was conducted in four-point bending, as illustrated in Figure 6. The results of these tests are shown comparatively in the fatigue stress–number of cycles to failure (S–N) diagram shown in Figure 11. The individual grouping of samples is illustrated in Figure 11 by light color regimes: EOS nominal, P3 keyhole, P5 process window, P8 lack of fusion, and EOS nominal improved. The poorest fatigue performance is observed for the P3 Keyhole specimens, having maximum fatigue stresses of ~ 950 MPa and a corresponding fatigue life between 10^5^ and 10^6^ cycles, slightly below the P5 process window. The P8 LoF samples failed just above 10^6^ cycles, while the best fatigue performance occurs for the EOS nominal improved samples near 10^7^ cycles. The surface roughness results are similar to the uniaxial fatigue tests of Cao, et al. [16] for various AM, as-built Ti64 alloy fatigue components. However, the S–N defect regions of Figure 11 differ from those of Narra, et al. [23], where the process window peak stress varied from 1200 MPa to ~980 MPa, corresponding to >10^7^ cycles to failure; nearly 10^2^ cycles better performance. In addition, the lack-of-fusion (LoF) 4-point bending test samples in the work of Narra, et al. [23] exhibited the poorest fatigue life of ~10^6^ cycles at peak fatigue stresses ranging from 400 to 500 MPa. In contrast, the Keyhole test specimens exhibited fatigue life. The 10^6^ cycles at peak stresses ranged from 600 to 500 MPa.

It is notable in the current study that the P3 keyhole specimens exhibited a pore circularity next highest to the EOS nominal specimens (0.66) but the highest circularity of 0.76 as well as the next largest pore diameter of 29 microns (Table 5). Table 6 also illustrates that while the EOS nominal improved specimens had the lowest surface roughness (S_a_ = 7 microns), they also had the lowest circularity of 0.57 and the smallest pore diameter of ~ 19 microns. Ye, et al. [19] showed that on plotting the mean fatigue life cycles versus maximum surface roughness (S_a_) for LPBF-fabricated Ti64, a S_a_ of ~7 microns corresponded to a mean fatigue life of just under 10^5^ cycles; nearly 10^2^ cycles lower than the EOS Nominal Improved components tested in Figure 11.

It is clear from examining Table 5 and Table 6 and Figure 11 that, as noted in numerous fatigue performance studies [26,27,28,29,30], pore size and shape (sphericity or circularity, as well as highly irregular, large pores), in combination with surface roughness conditions, influence both the maximum fatigue stress and the cycles to failure. This feature is also independent of the fatigue testing mode: uniaxial or multi-point bending. Indeed, Ye, et al. [19] show that surface roughness of <1 micron can only achieve a mean fatigue life of ~10^5^, as noted above. Sabota, et al. [52] also showed that spherical pores (having an equivalent circularity of 1) and a very small pore size promoted a higher fatigue life. This is due in part because there is a critical pore size below which there is no fatigue crack initiation [27]. Cao, et al. [16], among others, showed that only after post-processing of LPBF components (including HIP) could the fatigue life be extended to 10^9^, and at fatigue stresses of ~500 MPa.

### 4.4. S–N Curve Fitting with the Basquin Stress–Life Relation

While there have been numerous examples of S–N curve fitting using Basquin’s law of fatigue [2,3,8,54,55,56,57], including numerous AM alloy fatigue data [8], there are no examples of S–N data fitting for process window experiments for LPBF-fabricated Ti-6Al-4V; including the range of printing parameters as presented in Figure 11. The primary feature of Basquin’s law of fatigue [3] states that the lifetime of a component or part under fatigue increases as a power law when the external load amplitude decreases. This can be generally expressed as
S = AN^B^,(2)
where S is the maximum (peak) fatigue stress, N is the fatigue life, or number of cycles to failure, and A and B (the curve fitting parameters) are materials constants, although B depends on the damage accumulation.

Taking the log of (2) results in
log (S) = logA + Blog(N),(3)

A further common assumption is that for very low cycles (~10^3^), S can be approximated to be ~0.9 UTS, where in Figure 11, this value is ~900 MPa. Furthermore, at the endurance limit (S_e_), the peak stress below which there is no fatigue failure, S = S_e_ in Equation (3). Fitting these features to the process parameters in Figure 11, along with the corresponding N values, allows the fitting parameters, A and B, to be calculated for the specific parameter fitted curve. Figure 12 illustrates these results. In addition, these testing outcomes were compared with the mean discrepancy between the actual S–N test outcomes in Figure 11 and the fitted data, as shown in Table 6, which compares the fatigue stress level with the percent error by process parameter.

It is notable in Figure 12, as well as Table 7, that there is an extremely poor fitting for the keyhole process parameter while other parameters are better fit and more consistent. The keyhole parameter registered the shortest fatigue life; attributable to the pronounced surface roughness inherent in the keyhole process parameter, and an indication of the minor role played by internal defects in the reduction of fatigue life.

The preponderance of AM Ti-6Al-4V fatigue data is characterized by high-cycle, uniaxial fatigue testing and involves a very significant database established within the past few years [6,7,8,58,59,60]. A great deal of this data, especially fatigue life data, has been modeled or fitted to variations of Basquin’s exponential equation (Equation (2)), which has been variously modified for predicting fatigue parameters and performance [54,57,61]. In damage-tolerance design strategies, parts and structures experiencing fatigue loading must be able to sustain the inevitable incorporation of flaws and defects for a predicted useful life, which includes inspection periods after which the component can be repaired or replaced. For LPBF-fabricated Ti-6Al-4V, fatigue application and evaluation must be based predominantly on establishing a safe zone within a laser beam power level (P) and velocity (V) of the scanned beam in optimizing the fabrication of parts to be subjected to fatigue loading. In the present study, as well as the prior work of Narra, et al. [23], low-cycle, four-point bending fatigue tests for LPBF-Ti6Al-4V components have demonstrated the concept of a process window for as-built products where fatigue life can be qualitatively predicted based on the production of internal defects (especially LoF porosity) and surface roughness, which are examined by 2D optical metallography (Figure 7, Figure 8 and Figure 9) and scanning electron microscopy (SEM) (Figure 10), respectively. Table 5 shows that porosity can be well defined by process parameters, while Table 6 shows that, correspondingly, surface roughness also follows these process parameters, where it has been established generally that the average surface roughness parameter (S_a_) effectively qualifies LPBF-fabricated component surfaces. In this regard, surface quality strongly influences fatigue life.

The extreme deviation from Basquin’s law fitting the S–N curve of Figure 11 for internal keyhole porosity for the current process and printing parameters (Figure 12, Table 7) attests to the negligible influence of this defect, especially in light of the universality of the Basquin’s equation and its various forms [57]. This feature highlights the ability to utilize the process window of optimized printing parameters and process parameters and predict low-cycle, four-point bend fatigue life.

## 5. Conclusions

This study has comprehensively evaluated the fatigue performance of LPBF-as-built Ti-6Al-4V alloy components through four-point bending fatigue testing. The findings underscore the significant impact of surface finish and internal porosity (especially LoF) characteristics on the fatigue life of these components. Notably, the optimization of surface roughness to a S_a_ value of 7 microns from an initial 15 microns emerged as a pivotal factor in enhancing fatigue life performance, achieving ~10^7^ cycles to failure at a stress amplitude of ~300 MPa. This improvement highlights the critical role of surface conditions in fatigue resistance. A comparison of prominent surface roughness parameters also indicated that the average surface roughness, S_a_, provides an adequate representation of surface roughness.

Moreover, the present research delineated the complex interplay between surface roughness, pore circularity, and size, further elucidating their collective influence on the detrimental effects of keyhole and lack-of-fusion porosities on fatigue performance. Such intricacies point to the necessity of a nuanced understanding of material properties and defect morphology in predicting and improving the fatigue behavior of additive manufactured components.

Building on the foundations laid by previous studies [23], the present work reinforces the importance of careful manipulation of power–velocity (P–V) parameters within the designated process window of P–V space. By doing so, it is possible to significantly optimize the fatigue performance of as-built Ti-6Al-4V alloy components. A particularly notable feature of this work was the application of Basquin’s Law in the fitting of the S–N curve data corresponding to both printing and process parameters. While the Basquin stress–life relation fit the LoF and surface roughness process parameters, it was a very poor fit for the keyhole porosity, indicating that keyhole porosity plays a very small role in optimized LPBF-fabricated Ti-6Al-4V components where surface roughness dominates. This is an important feature that encompasses over 100 years of fatigue performance data ([3,8,36]) as it applies to complex components of L-PBF-fabricated Ti-6Al-4V alloy, in particular.

## Figures and Tables

**Figure 1 materials-17-04548-f001:**
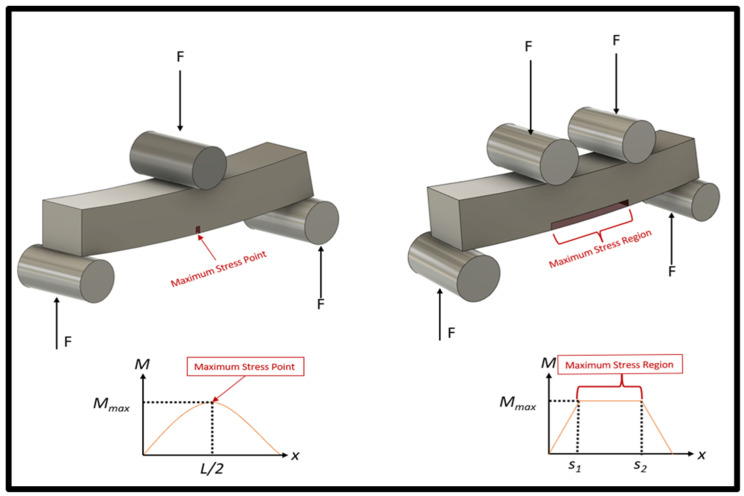
Comparison of the four-point bending and the three-point bending moment diagrams.

**Figure 2 materials-17-04548-f002:**
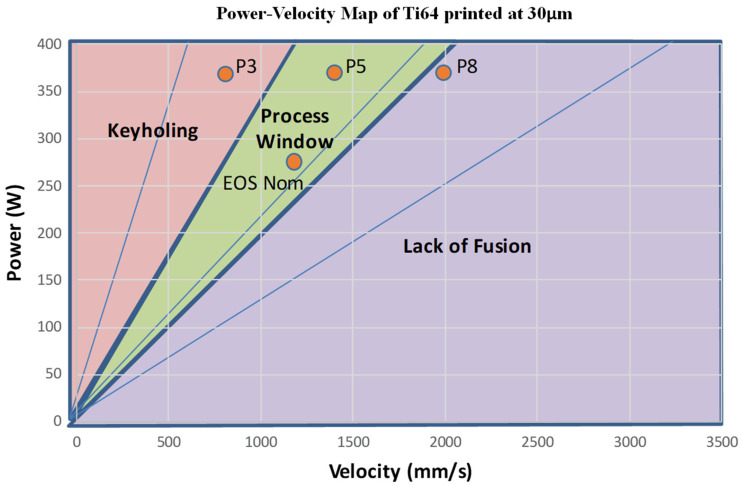
Process parameters selected for printing (orange).

**Figure 3 materials-17-04548-f003:**
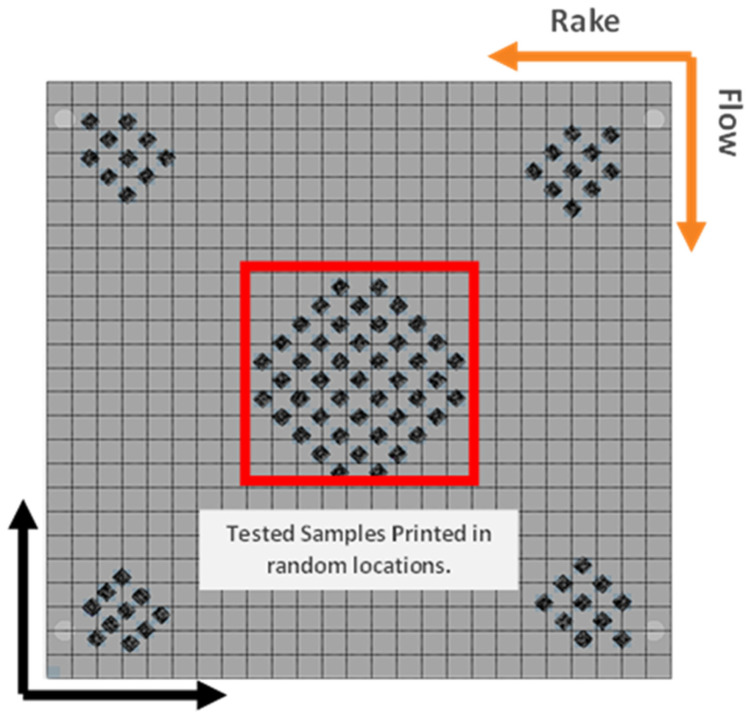
Print orientation of the tested samples (in the center) printed at randomly assigned process parameters.

**Figure 4 materials-17-04548-f004:**
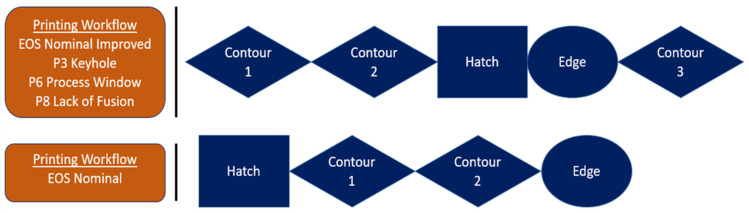
Scanning strategies used for each process parameter.

**Figure 5 materials-17-04548-f005:**
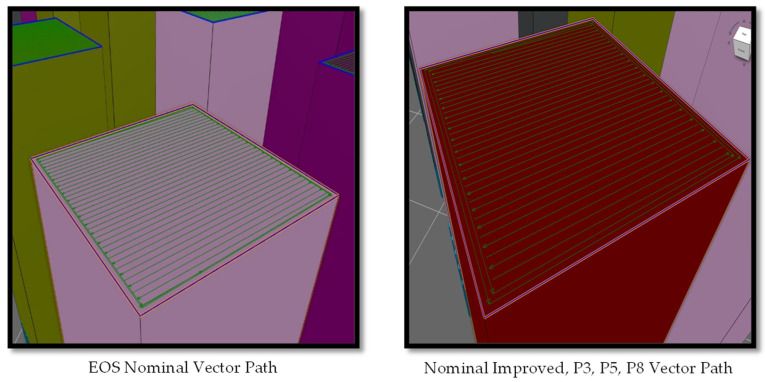
Scanning strategies used for each parameter.

**Figure 6 materials-17-04548-f006:**
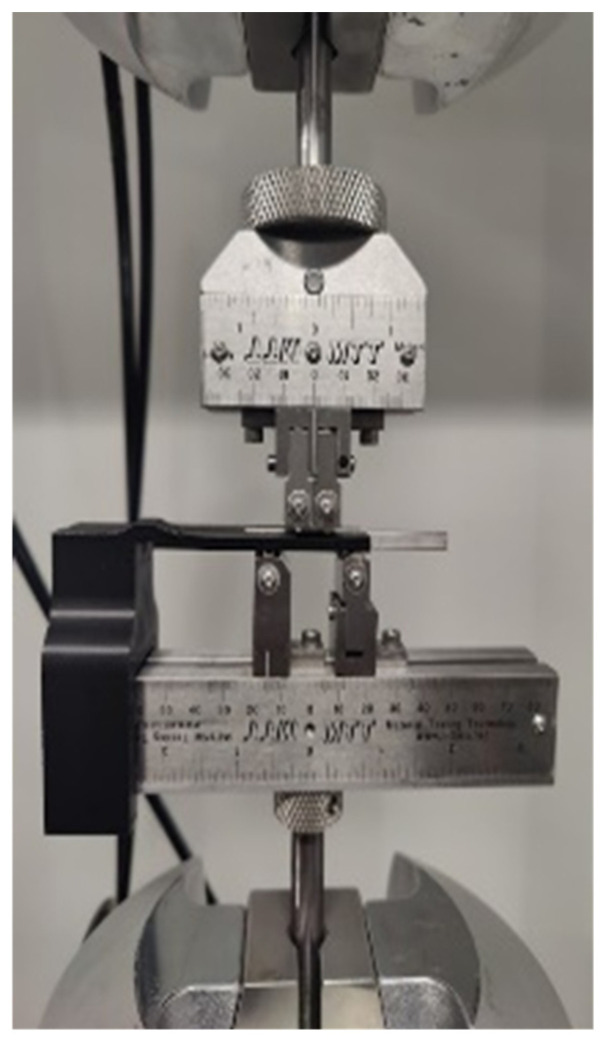
4-point bending fatigue test setup.

**Figure 7 materials-17-04548-f007:**
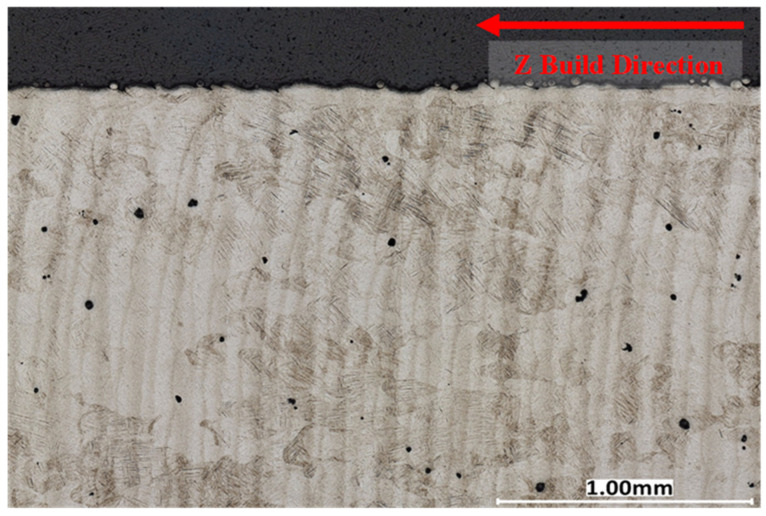
Ti-6Al-4V printed with keyholing parameters 370 W and 800 mm/s, 99.62% Dense. Taken using a Keyence VHX-7000.

**Figure 8 materials-17-04548-f008:**
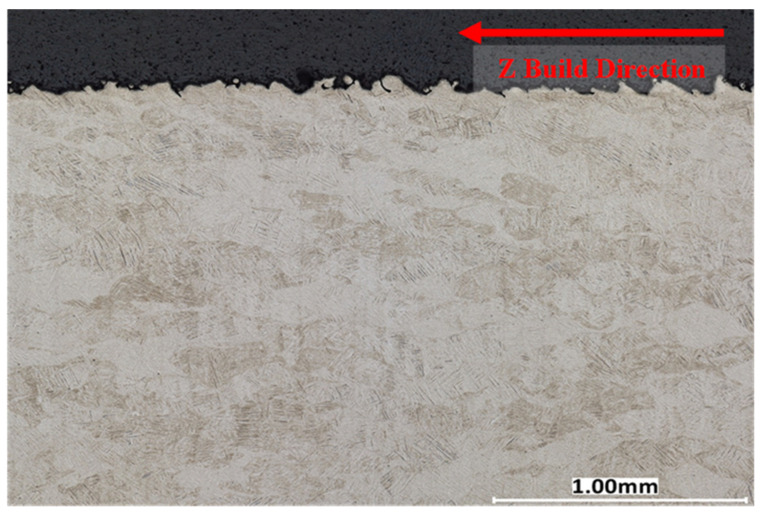
Ti-6Al-4V printed EOS Nominal Parameters. The parameters are 280 W and 1200 mm/s, 99.99% Dense. Taken using a Keyence VHX-7000.

**Figure 9 materials-17-04548-f009:**
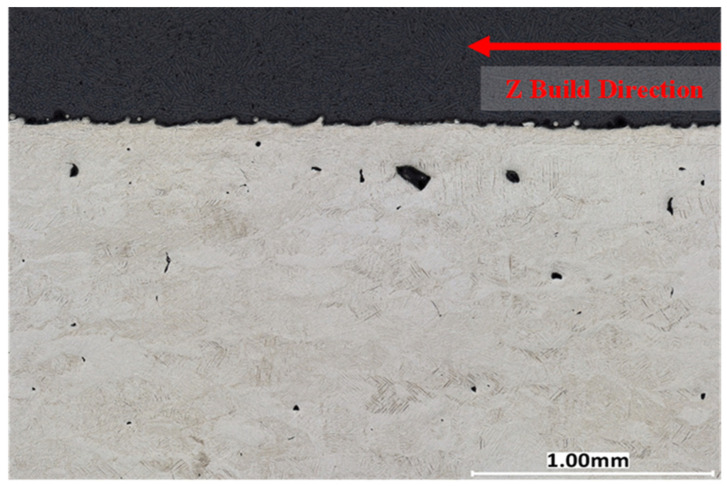
Ti-6Al-4V printed with lack of fusion parameters 370 W and 2000 mm/s, 99.79% Dense. Taken using a Keyence VHX-7000.

**Figure 10 materials-17-04548-f010:**
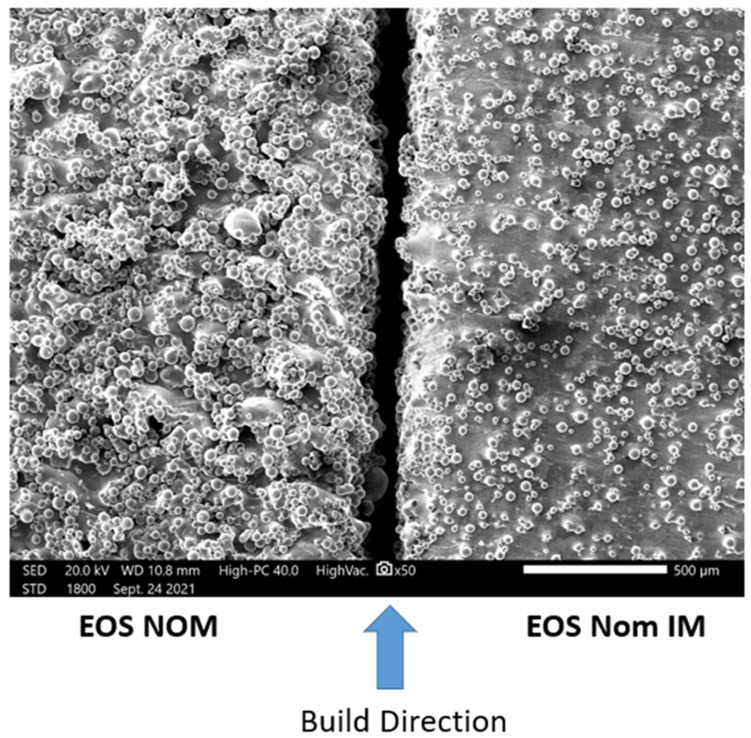
SEM surface comparison between EOS nominal (NOM) and EOS nominal improved.

**Figure 11 materials-17-04548-f011:**
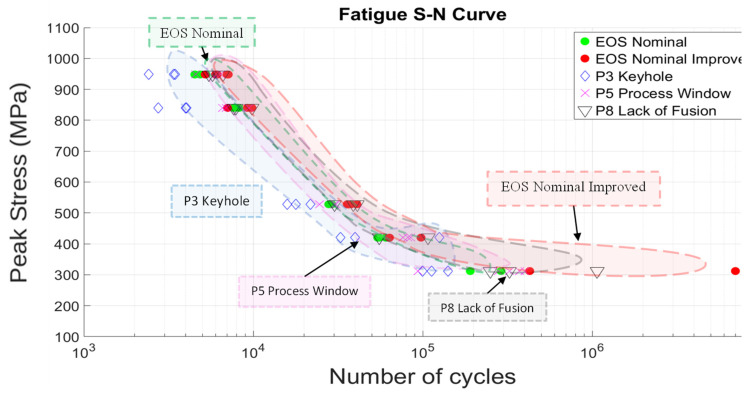
Fatigue peak stress vs. number of cycles to failure curves.

**Figure 12 materials-17-04548-f012:**
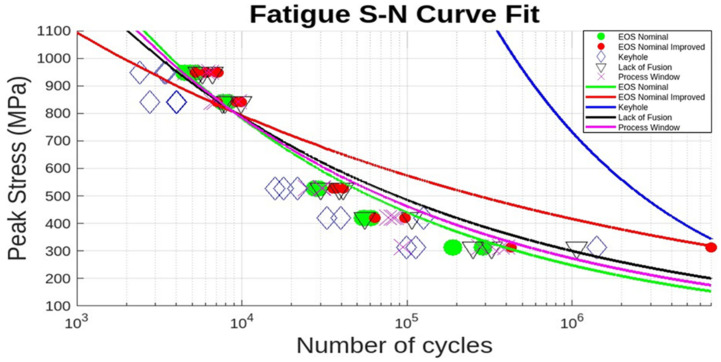
Basquin’s law fit of the S–N curves in Figure 11 (colored lines).

**Table 1 materials-17-04548-t001:** Powder chemical composition of Ti-6Al-4V provided by ATI (wt.%).

Alloy	Ti	Al	V	Fe	O	C	N	Y	H
Ti-6Al-4V	Balance	6.09	3.89	0.22	0.149	0.011	0.013	<9 × 10^−4^	8 × 10^−4^

**Table 2 materials-17-04548-t002:** Process parameter values.

Process Parameters	Laser Power (J)	Scanning Velocity (mm/s)
P3 Keyhole	370	800
P5 Process Window	370	1400
P8 Lack of Fusion	370	2000
EOS Nominal	280	1200
EOS Nominal Improved	280	1200

**Table 3 materials-17-04548-t003:** Contouring strategies for EOS nominal.

EOS Nominal Contour Strategy					
Contour 1 Offset (µm)	Contour 1 Power (W)	Contour 1 Scanning Speed (mm/s)	Contour 2 Offset (µm)	Contour 2 Power (W)	Contour 2 Scanning Speed (mm/s)
20	150	1250	0	150	1250

**Table 4 materials-17-04548-t004:** Contouring strategies for EOS nominal improved, P3, P5, P8.

EOS Nominal Improved, P3, P5, P8 Contour Strategy								
Contour 1 Offset (µm)	Contour 1 Power (W)	Contour 1 Scanning Speed (mm/s)	Contour 2 Offset (µm)	Contour 2 Power (W)	Contour 2 Scanning Speed (mm/s)	Contour 3 Offset (µm)	Contour 3 Power (W)	Contour 3 Scanning Speed (mm/s)
0	100	450	80	100	450	0	100	550

**Table 5 materials-17-04548-t005:** Values of defect geometry, porosity, number of defects, and average diameter on a cross-section of the specimen categorized by process parameter.

Printing Parameter	Power (W)	Scanning Speed (mm/s)	% Porosity	Circularity Avg.	Number of Defects	Avg. dia. (µm)
P3 Keyhole	370	800	0.385 ± 0.004	0.76 ± 0.04	3861	29
P5 Process Window	370	1400	0.106 ± 0.012	0.63 ± 0.06	1363	32
P8 Lack of Fusion	370	2000	0.215 ± 0.046	0.58 ± 0.05	2708	26
EOS Nominal	280	1200	0.008 ± 0.004	0.66 ± 0.06	231	22
EOS Nominal Improved	280	1200	0.017 ± 0.012	0.57 ± 0.05	143	19

**Table 6 materials-17-04548-t006:** Process parameter versus average surface roughness (S_a_), maximum valley depth (S_v_), and average valley depth (S_vk_).

Process Parameter	S_a_ (µm)	S_v_ (µm)	S_vk_ (µm)
EOS Nominal	15	122	17
EOS Nominal Improved	7	92	9
Keyhole	12	78	15
Lack of Fusion	8	48	8
Process Window	9	64	11

**Table 7 materials-17-04548-t007:** Percentage error of Basquin’s law fit to S–N process parameter data in Figure 11 (Figure 12).

Process Error by Process Parameter					
Stress Level Tested	NOM	NOM IM	Keyhole	Lof	PW
312 MPa	9%	17%	449%	11%	8%
420 MPa	17%	43%	387%	23%	13%
528 Mpa	12%	26%	539%	15%	15%
840 MPa	−3%	−3%	664%	−2%	−2%
946 MPa	−3%	−10%	618%	−6%	−7%
Average % Error	6%	15%	532%	8%	5%

## Data Availability

Data are available upon request.

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
