# Peer review of "Effects of Process Parameters and Process Defects on the Flexural Fatigue Life of Ti-6Al-4V Fabricated by Laser Powder Bed Fusion"

_materials, 2024, doi:10.3390/ma17184548_

Round 1
Reviewer 1 Report
Comments and Suggestions for Authors
The article focused on the fatigue properties of LPBF Ti64 alloy. Ti64 specimens were fabricated using different LPBF parameters and subsequently tested in 4-point bending testing.
The objective of the article was to provide “an extension to the recent work of Narra, et al [23] on LPBF-fabricated Ti64 alloy, while also complementing the prior work of Cao, et al. [16]”. They indicated that “the effect of surface roughness (Sa, Sv and Svk) was investigated and compared with the porosity-related fatigue performance issues and process parameters by plotting the corresponding peak fatigue stress (S) against the number of cycles to failure(N) in S-N diagrams”.
However, the methodology section is incomplete and does not provide clarity regarding how the specimens were printed, the post-treatment procedures and the specifics of the measurement methds used. For example:
1. The authors presented a power-velocity map and referenced the work of Jerard V. Gordon. The map is only a prediction based on a limited number of experimental results. However, the authors used this map to determine printing parameters for their study and to print specimens containing keyhole and LoF defects. Moreover, the authors used different printing parameters than those in [50], and details such as hatching space and scanning strategy (rotation between layers) were not specified. The lack of information makes it difficult to undestand how the printing parameters were chosen and whether they actually correspond to the printing windows that lead to keyhole and LoF defects.
2. How were the specimens for the bending test printed? What was their orientation? It is very important information, especially if you are producing LoF defects, as it helps to understand how their shape can affect the fatigue results.
3. Simply mentioning ASTM standards is insufficient when you present post-heat treatment . It's important to specify the temperature, duration, and atmosphere used for stress-relief treatment. These parameters significantly influence the fatigue results.
4. Why was surface roughness measured using optical imaging? If the goal is to evaluate the effect of surface finishing on fatigue properties, a standardized test method must be used. It is important to clarify which specific surface was analyzed, because the surface roughness of LPBF specimens orientation dependent.
5. The same is for porosity: Optical images were presented for only 3 out of 5 specimens, and it seems that they were observed close to the surface? How does it correspond to the volumetric representation of porosity. Why was porosity measured only in the XY cross-section? How does this relate to the axis of applied fatigue stress?
6. In the fatigue test, how was the maximum bending stress chosen and what was the minimum stress? What was the procedure for specimen preparation before the fatigue test, especially the removal of support and surface? Did the specimen dimensions meet the standards for this type of test? Why was one sample subjected to two tests at different stress levels? Please provide more information?
Without this information, it is difficult to judge the quality and reliability of the results and conclusions of the study.
For example, the authors wrote : “It is notable in the current study that the P3 Keyhole specimens exhibited a surface roughness next highest to the EOS Nominal specimens (0.66), but the highest circularity of 0.76 as well as the largest pore diameter of 29 microns (Table 5).” However, according to Table 5, the largest pore diameter is attributed to P5, note P3. Irregular pores generally have a more detrimental effect on fatigue properties, which contrasts with the presented findings. How can you explain this?
The authors used “number of pores” as a characteristic of pores. Is this number based on a specific cross-section? How does it relate to the overall pore distribution and volume fraction in the entire specimen? Detailed analysis of pore volume distribution, shape, and size in relation to fatigue stress axes is important to study the contribution of LoF and keyhole defects.
The microstructure of Ti64 processed with different printing parameters significantly influences fatigue properties but it is not observed in this study.
The same is for fracture surface analysis. Without such analysis, it's difficult to conclude how these factors (surface, pores and microstructure) impacted fatigue results and what was the contribution of each of them.
General remarks:
1. Please review Table 1, it contains empty lines
2. Section 3.3 starts with a table without introduction
3. Clarify the labeling of specimens: P5 in Figure 2 and P6 in Table 2. Why numbering strats with 3?
4. Please organize Figures 7, 8 and 9 together for direct comparison. Where are other specimens, please include images for all specimens?
Author Response
Dear Editor/Reviewer,
We have carefully considered the extensive comments and recommendations for our revision of our paper, “Effects of Process Parameters and Process Defects on the Flexural Fatigue Life of Ti-6Al-4V Fabricated by Laser Powder Bed Fusion” (materials-3105465), which have been valuable in clarifying the enclosed revision. We have addressed the comments as follows:
Comments 1 and 2: Figures 2-5 along with tables 2-4 illustrate the process parameters used to build the specimens for this experiment. Figures 7 and 9 demonstrate that keyhole and lack of fusion were achieved, respectively.
Comment 3: This sentence has been updated as recommended… To ensure residual stresses did not affect the fatigue life of the samples, the forty-five samples were stress-relieved at 600°C for 120 minutes in vacuum.
Comment 4: Optical method for surface roughness was chosen because it allowed the opportunity to analyze the entirety of the testing region and produce a representative value of that said region. Paragraph 2 in section 4.2 clarifies that the reported roughness values correspond to the tension-side of the specimen i.e. the region highlighted in figure 1.
Comment 5: The porosity was analyzed optically to qualify whether the desired porosity was present. The porosity prevailed mostly near the surface, and this was of greater importance due to the nature of the fatigue testing.
Comment 6: The stress ratio has been added to the methods section. The fatigue testing was performed on as-built specimens following stress relief; the specimens were not machined. The discrepancy regarding the discussion of results has been corrected.
General Remarks: The issues have been addressed.
We have also gone through the English language features of the revision to assure good compliance. We hope this revision will meet the high standards of the journal and look forward to its acceptance.
Respectfully,
Cristian Banuelos
Reviewer 2 Report
Comments and Suggestions for Authors
This manuscript systematically studied the fatigue performance of laser powder-bed fusion fabricated (LPBF) Ti-6Al-4V alloy by using 4-point bending fatigue testing. The results highlight the importance of surface finish and internal porosity on the fatigue life of the components. They also conclude that the keyhole porosity has relatively small contribution to the LPBF-fabricated Ti-6Al-4V alloy components. Here are my comments:
1. In page 3 line 107, you mentioned this work is based on the recent work by Narra and Cao. Compared to previous work, what are the specific characters and advantages of using the 4-point bending test method? Did this method yield different conclusions from previous methods?
2. Has anyone used this 4-point bending fatigue testing? Please add more reference for this.
3. In Table1, why there are two empty lines ?
4. What is the difference between the data in Figure 11 and Figure 12( EOS nominal, EOS Nominal improved, P3 Keyhole, P5 Keyhole, P8 lack of Fusion). And figure 11 is not complete.
Author Response
Dear Editor/Reviewer,
We have carefully considered the comments and recommendations for our revision of our paper, “Effects of Process Parameters and Process Defects on the Flexural Fatigue Life of Ti-6Al-4V Fabricated by Laser Powder Bed Fusion” (materials-3105465), which have been valuable in clarifying the enclosed revision. We have addressed the comments as follows:
Comment 1: The use of 4-point bend testing or multiaxial fatigue testing, especially for additively manufactured alloys such as Ti-64 is especially applicable to complex components fabricated by AM. While the results of performance are not much different from uniaxial testing, multiaxial testing is more representative of the multiaxial stress states experienced by complex geometries. References [23] nu Narra, et al. and especially the extensive review by Foti, et al. [36] illustrates these issues in detail; while the review by Zhang and Wu [8] examines and compares fatigue performance data for more than 3000 AM publications. We have also very recently compared multiaxial and uniaxial fatigue performance data for Ti-64, and we add this paper as reference [37] in the revision.
Comment 2: As we note in the comment above, many publications have used 4-point bending fatigue testing as we describe. Reference [37] has been added as noted also.
Comment 3: This has been resolved.
Comment 4: Figure 11 is the raw fatigue data expressed Stress vs. Cycles to failure graph while figure 12 takes this data and demonstrates how the data fits Basquin’s Law.
We have also gone through the English language features of the revision to assure good compliance. We hope this revision will meet the high standards of the journal and look forward to its acceptance.
Respectfully,
Cristian Banuelos
Reviewer 3 Report
Comments and Suggestions for Authors
This manuscript presents a comprehensive study on the effects of process parameters and defects on the flexural fatigue life of Ti-6Al-4V fabricated by laser powder bed fusion (LPBF). The authors have conducted a thorough investigation using various analytical techniques and fatigue testing methods. Overall, the work is well-structured and provides valuable insights into the fatigue performance of LPBF-fabricated Ti-6Al-4V components. However, there are several areas where the manuscript could be improved:
- The introduction provides a good overview of the field, but it could be strengthened by including more recent literature (2022-2023) on fatigue performance of LPBF-fabricated Ti-6Al-4V. This would help to better contextualize the novelty of the present work.
- In the experimental section, more details should be provided on the LPBF process parameters, such as layer thickness, hatch spacing, and scanning strategy. These parameters can significantly influence the microstructure and defect formation.
- The authors mention using a simplified power-velocity (P-V) map corresponding to 30-micron layer thickness. It would be beneficial to include this map in the manuscript and discuss how it was developed.
- The discussion of porosity analysis could be expanded. The authors should consider quantifying the porosity levels for each process parameter and discussing how these relate to the fatigue performance.
- In Figure 11, the S-N curves for different process parameters are presented. It would be helpful to include error bars to show the scatter in the fatigue life data.
- The Basquin's law fitting section is interesting, but the authors should discuss the physical meaning of the fitted parameters (A and B) and how they relate to the material properties and process parameters.
- The conclusions could be strengthened by more explicitly stating the novelty and significance of the findings in the context of LPBF-fabricated Ti-6Al-4V components.
- Throughout the manuscript, there are minor grammatical and typographical errors that should be corrected. For example, in line 39, "Partial Dischargers" should be "Partial Discharges".
- The authors should consider adding a brief discussion on the practical implications of their findings for the design and manufacturing of LPBF Ti-6Al-4V components.
- The environmental implications of using LPBF for Ti-6Al-4V fabrication should be addressed, perhaps in the introduction or conclusions.
Moderate editing of English language required
Author Response
Dear Editor/Reviewer,
We have carefully considered the extensive comments and recommendations for our revision of our paper, “Effects of Process Parameters and Process Defects on the Flexural Fatigue Life of Ti-6Al-4V Fabricated by Laser Powder Bed Fusion” (materials-3105465), which have been valuable in clarifying the enclosed revision. We have addressed the comments as follows:
Comment 1: Reference [8] compares more than 3000 publications dealing with the wide range of fatigue performance data for AM up to 2023. Reference [36] by Foti, et al. also provides an extensive review of multiaxial fatigue testing of AM-fabricated alloys, and especially describes the consideration of multiaxial stress states in complex AM components. A recent paper by our group comparing 4-point and uniaxial loading of Ti-64 (Ariza, et al, J. Mater. Res. & Technol., 2024) has been added as reference [37] along with a very recent paper by Rogers, et al. added as reference [38]. A narrative discussing these additions has been added to the Introduction (paragraph 4): …Additionally, Foti, et al. [36] have noted that complex, AM-fabricated parts or systems often experience multiaxial (flexural) fatigue as a consequence of multiaxial stress states. Correspondingly, Ariza, et al. [37] have also recently compared the 4-point and uniaxial fatigue performance for L-PBF-fabricated Ti-6Al-4V alloy. Rogers, et al. [38] recently described methodologies for quantifying AM fatigue data uncertainty.
[37] D. A. Ariza, E. Arrieta, C. Banuelos, B. J. Colon, L. E. Murr, R. B. Wicker, C. Beamer, F. Medina, Comparison of fatigue like behavior between 4-point and uniaxial loading for L-PBF Ti-6Al-4V after HIP treatments. Results in Mater. 22(2024) 100579.
[38] J. Rogers, M. Gian, J. Elambasseril, C. Burvill, C. Brice, C. Wallbrink, M. Brandt, M. Leary, Fatigue test date applicability for additive manufacture: A method for quantifying the uncertainty of AM fatigue data. Mater. Design 231 (2023) 111978.
Comment 2: Figures 2-5 along with tables 2-4 illustrate the process parameters used to build the specimens for this experiment.
Comment 3: This map has been included as Figure 2.
Comment 4: The presence of porosity was observed in this study however it was determined that surface roughness had a much more significant effect on the fatigue performance.
Comment 5: The addition of error bars to figure 11 may hinder the ability to read the graph given how close the data points are to each other. Table 7, however, highlights this error for the reader for each process parameter at each testing stress level.
Comment 6: Regarding Basquin’s fitting equation, the materials constants A and B are briefly discussed; intimating that B is specifically related to damage accumulation. Eq. (3) illustrates the context of A and B as they relate to the fatigue stress, S. The text describes how A and B are calculated. The references, [55-58] provide additional background. It can be observed that A = Se/106B; where Se is the endurance limit for the S-N data. It would not seem appropriate to go through a detailed calculation of A and B in the body of the paper, and reference [4] provides some context.
Comment 7: As suggested, we have added the following sentence to the last paragraph of the manuscript…..where surface roughness dominates. This is an important feature which encompasses over 100 years of fatigue performance data ([3], [8] and [36]) as it applies to complex components of L-PBF-fabricated Ti-6Al-4V alloy in particular.
Comment 8: This issue has been addressed.
We have also gone through the English language features of the revision to assure good compliance. We hope this revision will meet the high standards of the journal and look forward to its acceptance.
Respectfully,
Cristian Banuelos
Round 2
Reviewer 2 Report
Comments and Suggestions for Authors
The revised manuscript and author's reply has answered all the questions.
Reviewer 3 Report
Comments and Suggestions for Authors
Accept